# MTAP deficiency creates an exploitable target for antifolate therapy in 9p21-loss cancers

Omar Alhalabi [1,12], Jianfeng Chen[1,12], Yuxue Zhang[1,12], Yang Lu [2,12], Qi Wang [3], Sumankalai Ramachandran[1], Rebecca Slack Tidwell [4], Guangchun Han [5], Xinmiao Yan[5], Jieru Meng[1], Ruiping Wang[5], Anh G. Hoang[1], Wei-Lien Wang [1], Jian Song[1], Lidia Lopez[1], Alex Andreev-Drakhlin[1], Arlene Siefker-Radtke [1], Xinqiao Zhang[1], William F. Benedict[1], Amishi Y. Shah [1], Jennifer Wang[1], Pavlos Msaouel[1], Miao Zhang[6], Charles C. Guo[6], Bogdan Czerniak[6], Carmen Behrens[7], Luisa Soto [8], Vassiliki Papadimitrakopoulou[7], Jeff Lewis[4], Waree Rinsurongkawong[4], Vadeerat Rinsurongkawong[4], Jack Lee [4], Jack Roth [9], Stephen Swisher [9], Ignacio Wistuba[6], John Heymach [7], Jing Wang [3], Matthew T. Campbell [1], Eleni Efstathiou[1], Mark Titus[1], Christopher J. Logothetis [1], Thai H. Ho [10,13], Jianjun Zhang [7,13], Linghua Wang [5,11,13 ✉] & Jianjun Gao [1,13 ✉]

Methylthioadenosine phosphorylase, an essential enzyme for the adenine salvage pathway, is often deficient (MTAP^def) in tumors with 9p21 loss and hypothetically renders tumors susceptible to synthetic lethality by antifolates targeting de novo purine synthesis. Here we report our single arm phase II trial (NCT02693717) that assesses pemetrexed in MTAP^def urothelial carcinoma (UC) with the primary endpoint of overall response rate (ORR). Three of 7 enrolled MTAP^def patients show response to pemetrexed (ORR 43%). Furthermore, a historic cohort shows 4 of 4 MTAP^def patients respond to pemetrexed as compared to 1 of 10 MTAP-proficient patients. In vitro and in vivo preclinical data using UC cell lines demonstrate increased sensitivity to pemetrexed by inducing DNA damage, and distorting nucleotide pools. In addition, MTAP-knockdown increases sensitivity to pemetrexed. Furthermore, in a lung adenocarcinoma retrospective cohort (N = 72) from the published BATTLE2 clinical trial (NCT01248247), MTAP^def associates with an improved response rate to pemetrexed. Our data demonstrate a synthetic lethal interaction between MTAP^def and de novo purine inhibition, which represents a promising therapeutic strategy for larger prospective trials.

---

[1] Department of Genitourinary Medical Oncology, The University of Texas MD Anderson Cancer Center, Houston, TX 77030, USA. [2] Department of Nuclear Medicine, The University of Texas MD Anderson Cancer Center, Houston, TX 77030, USA. [3] Department of Bioinformatics and Computational Biology, The University of Texas MD Anderson Cancer Center, Houston, TX 77030, USA. [4] Department of Biostatistics,, The University of Texas MD Anderson Cancer Center, Houston, TX 77030, USA. [5] Department of Genomic Medicine, The University of Texas MD Anderson Cancer Center, Houston, TX 77030, USA. [6] Department of Pathology, The University of Texas MD Anderson Cancer Center, Houston, TX 77030, USA. [7] Department of Thoracic, Head and Neck Medical Oncology, The University of Texas MD Anderson Cancer Center, Houston, TX 77030, USA. [8] Department of Translational molecular pathology, The University of Texas MD Anderson Cancer Center, Houston, TX 77030, USA. [9] Department of Thoracic and Cardiovascular surgery, The University of Texas MD Anderson Cancer Center, Houston, TX 77030, USA. [10] Division of Medical Oncology, Mayo Clinic, Phoenix, AZ, USA. [11] The University of Texas MD Anderson Cancer Center UTHealth Graduate School of Biomedical Sciences (GSBS), Houston, TX, USA. [12] These authors contributed equally: Omar Alhalabi, Jianfeng Chen, Yuxue Zhang, Yang Lu. [13] These authors jointly supervised this work: Thai H. Ho, Jianjun Zhang, Linghua Wang, Jianjun Gao. ✉email: LWang22@mdanderson.org; jgao1@mdanderson.org

Despite major advances in immune checkpoint therapy (ICT) and targeted therapy[1–9], urothelial carcinoma (UC) remains a major cause of cancer-related deaths across the world and there continues to be an urgent need to better characterize and target distinct molecular subtypes of this disease[10–12]. Loss of the focal chromosomal region 9p21.3 is a common event in UC[13] that leads to loss of tumor suppressor genes CDKN2A/B and metabolic gene MTAP (Methylthioadenosine phosphorylase)[14,15] and correlates with adverse clinical outcomes[16–22]. The MTAP protein is an essential enzyme that controls the salvage synthesis of adenine from the substrate methylthioadenosine (MTA) (Fig. 1a)[14,15,19,23–25].

Adenine nucleotides are synthesized either by de novo biosynthesis, which is supported by folate-mediated single-carbon metabolism, or by salvage biosynthesis controlled by MTAP[24–27]. Therefore, we hypothesize that MTAP loss produces a synthetic lethal vulnerability to de novo adenine synthesis inhibition thus rendering MTAP-deficient (MTAP$^{def}$) tumor cells susceptible to antifolate agents (Fig. 1a) including pemetrexed, which is a potent antifolate agent that suppresses de novo purine synthesis (as well as thymidine synthesis) by inhibiting three of the key enzymes involved in folate metabolism, namely dihydrofolate reductase, thymidylate synthase, and glycinamide ribonucleotide formyl transferase[28]. As such, tumor MTAP deficiency may provide a metabolic vulnerability for the use of antifolate agents such as pemetrexed to effectively treat UC with 9p21 loss.

In this work, we test our hypothesis and analyze tumor tissues for MTAP expression status in correlation with clinical responses to pemetrexed in patients with metastatic UC who either enrolled on our phase II clinical trial (NCT02693717) or were historically treated with pemetrexed. We further corroborate our findings mechanistically using preclinical models and clinically using an independent clinical cohort of patients with metastatic lung adenocarcinoma.

## Results

**MTAP loss is prevalent in UC.** Based upon genomic data[13], more than 99% of MTAP$^{def}$ UC also contains CDKN2A loss. Since we already established a CLIA certified immunohistochemistry (IHC) test to determine MTAP gene expression status, we decided to use MTAP$^{def}$ as a surrogate biomarker for 9p21 loss. To evaluate the prevalence of MTAP loss in UC, we assessed The Cancer Genome Atlas (TCGA) database for MTAP somatic copy number alterations (SCNAs) and found that 106 of 408 (26%) muscle-invasive UC cases harbored a homozygous deletion of the MTAP gene[29] (Fig. 1b). Consistent with TCGA data, IHC analysis of a tumor tissue microarray from 151 UC patients demonstrated that MTAP loss occurred in 27.8% of tumor specimens (Fig. 1c). Of note, we observed no significant difference in the rate of MTAP loss between early-stage and advanced-stage tumors suggesting an early onset of this genomic event (Fig. 1d and Supplementary Table 1).

**Patient characteristics.** In a cohort of 21 patients, including a historical cohort of 14 metastatic UC patients (4 with MTAP$^{def}$ and 10 with MTAP$^{prof}$, respectively) treated with pemetrexed beyond first line of therapy (Supplementary Fig. 1) and our current trial cohort of 7 patients (all with MTAP$^{def}$) enrolled in clinical trial NCT02693717, the median age was 70 (Table 1). Approximately two thirds of the patients were male. More than half (52%) of the patients had a limited Eastern Cooperative Oncology Group (ECOG) performance status (1 or 2). Prior cisplatin-based chemotherapy was given in 16 of 21 (76%) of patients and prior ICT was given in 6 of 21 (28%) of patients. Patients enrolled in NCT02693717 (09/2017–2/2019) were more

likely to have received prior ICT (71%) as compared to patients included in our historical cohort (7%) as patients in the historical cohort were treated before the approval of ICT for UC (9/2014–2/2016). Furthermore, 47% of patients in our entire cohort had received ≥2 lines of prior therapy, as detailed in Table 1.

**Pemetrexed is clinically effective against MTAP$^{def}$ UC.** In our historical cohort analysis, all MTAP$^{def}$ UC patients (4 of 4) had an objective response to pemetrexed (ORR 100%), while only one of the 10 MTAP-proficient (MTAP$^{prof}$) UC had an objective response (ORR 10%). Notably, 8/10 patients with MTAP$^{prof}$ UC had an increase in tumor volume from baseline as best response (Fig. 1e).

Our prospective study NCT02693717 (Fig. 1f) enrolled 7 patients with previously treated MTAP$^{def}$ UC but was intentionally terminated early in favor of another trial with pemetrexed plus anti-PD-L1 combination therapy in the same population of patients. The primary endpoint was overall response rate (ORR) to pemetrexed, which was 42% (3 of 7 patients) (Fig. 1g). The secondary endpoints were survival and safety of pemetrexed. After a median follow up of 7.1 months, the estimated median progression-free survival (PFS) is 5.3 months and median overall survival (OS) is 7.7 months. All and treatment-related adverse events (AEs) experienced by patients treated with pemetrexed in our phase II trial are summarized in Supplementary Tables 2 and 3. There were no grade 4 or 5 treatment-related AEs. Most common treatment-related AE was anemia and includes grade 3 anemia in 3 of 7 (43%) of patients. Figure 1h shows an imaging example of a patient with metastatic MTAP$^{def}$ UC who had a near-complete response to pemetrexed (after progressing through first-line gemcitabine and cisplatin) as compared to a patient with metastatic MTAP$^{prof}$ UC who progressed on pemetrexed (after progressing through first-line gemcitabine and cisplatin).

**Antifolates are effective in human MTAP$^{def}$ UC cell lines and xenograft models by distorting nucleotide pools and inducing DNA double-strand breaks.** To further test our hypothesis that MTAP deficiency may create a vulnerability to antifolate agents, we assessed the sensitivity of UC cell lines to pemetrexed. For this purpose, we screened the available human UC cell lines, confirmed MTAP expression status using Western blot, and identified four MTAP$^{prof}$ UC cell lines (HT-1376, T24, HT-1197, and J82) and four MTAP$^{def}$ UC cell lines (RT112, RT4, UM-UC-3, and 253 J) (Fig. 2a). The T24 cell line was identified as MTAP$^{prof}$ but p16 (product of CDKN2A) deficient (Supplementary Fig. 2a). We also confirmed the functional consequences of MTAP loss by detecting a >30-fold increase in the concentrations of its substrate, MTA, in the culture medium of MTAP$^{def}$ UC cell lines as compared to the culture medium of MTAP$^{prof}$ UC cell lines (Fig. 2a and Supplementary Fig. 2b), consistent with previous reports[14,15,30]. We found that the viability of MTAP$^{def}$ UC cell lines in response to pemetrexed treatment was much lower (IC$_{50}$ < 0.25 μM) as compared to MTAP$^{prof}$ UC cell lines (IC$_{50}$ > 20 μM) (Fig. 2b). In addition, we ruled out that accumulating MTA has any impact on the MTAP$^{def}$ and MTAP$^{prof}$ cell lines viability in the absence or presence of pemetrexed (Supplementary Fig. 3a, b). On the other hand, we did not observe a difference in the viability of MTAP$^{def}$ cell lines in comparison to MTAP$^{prof}$ cell lines when treated with the antimetabolite gemcitabine, which is commonly used in urothelial cancer (Supplementary Fig. 4a). Notably, the T24 cell line did not demonstrate high levels of sensitivity to pemetrexed (Fig. 2b) despite being p16 deficient, suggesting that loss of p16 alone is not associated with increased sensitivity to pemetrexed and that the increased sensitivity is more likely attributed to MTAP loss. Consistent with

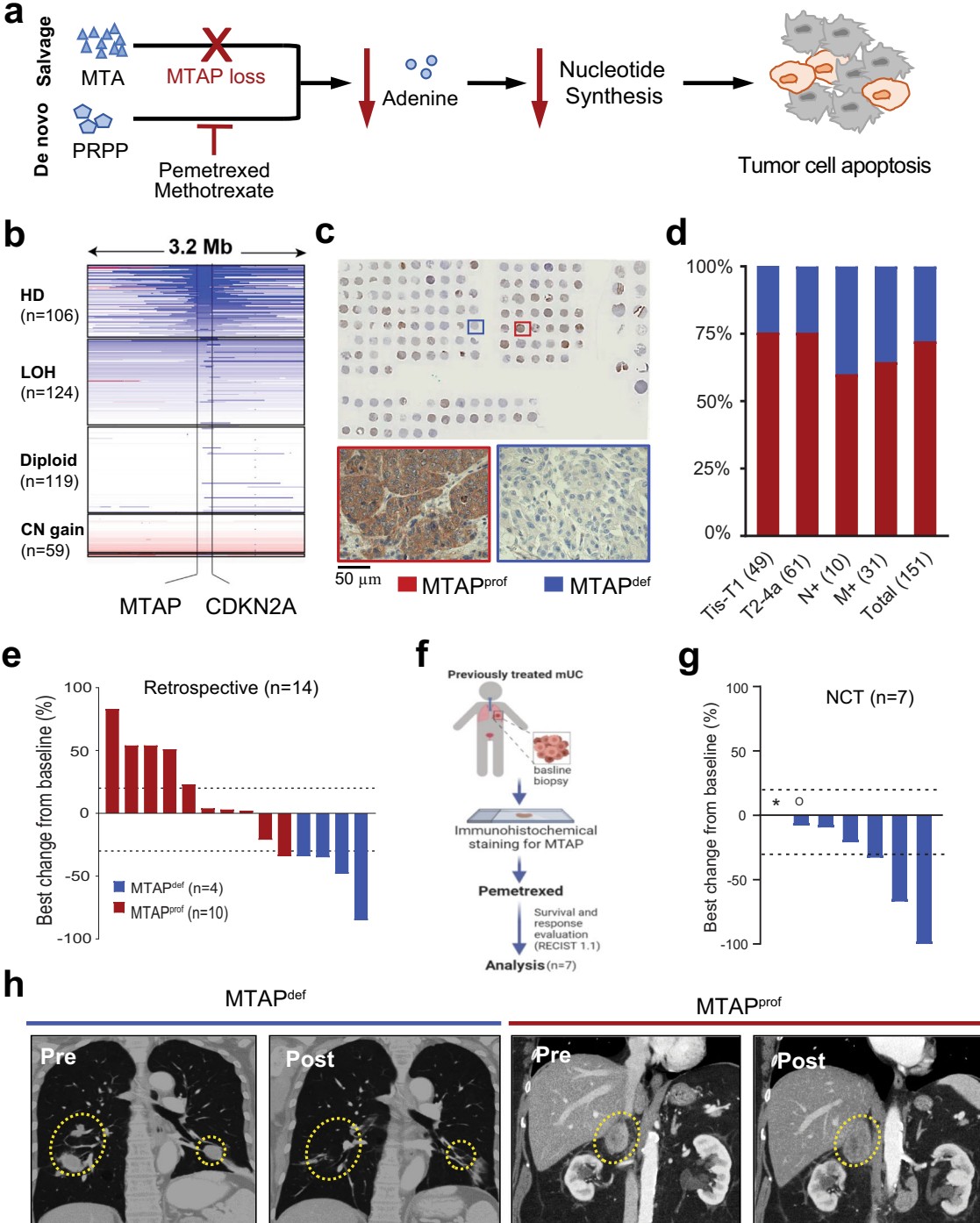

**Fig. 1 MTAP deficient metastatic urothelial carcinoma response to pemetrexed. a** schematic illustration of salvage and de novo adenine synthesis pathways in the context of MTAP loss and de novo purine synthesis inhibition. Inhibition of the de novo pathway leads to decreased nucleotide synthesis and eventually tumor cell apoptosis. MTA methylthioadenosine, PRPP phosphoribosyl pyrophosphate. **b** Frequency of *MTAP* deletion in 408 UC patients in The Cancer Genome Atlas (TCGA) database. HD, homozygous deletion; LOH, loss of heterozygosity; LLG, low-level copy number gain; Ampl, amplification. **c, d** Frequency of MTAP loss in a tumor tissue microarray. Positive and negative staining of MTAP within 109 MTAP^prof and 42 MTAP^def samples are available in Source Data file. Tis, carcinoma in situ; T1, invasive to lamina propria; T2-4a, muscle-invasive carcinoma; N +, metastatic to nodes; M +, systemic metastases. **e** Waterfall plots for best response of target lesions based on retrospective analysis of patients with metastatic UC treated with pemetrexed. **f** Clinical trial schema of NCT02693717. **g** Waterfall plots for best response of target lesions of patients with metastatic UC treated with pemetrexed under NCT02693717. *not evaluable for response; °progressive disease for new lesions. **h** Example of a patient with metastatic MTAP^def UC who progressed after receiving gemcitabine/cisplatin but responded to pemetrexed, compared to patient with MTAP^prof UC who progressed after receiving gemcitabine/cisplatin but didn't respond to pemetrexed.

**Table 1 Baseline patient characteristics of patient with metastatic urothelial cancer at the start of pemetrexed treatment.**

| | | Historic Cohort | | NCT02693717 | Total cohort |
|---|---|---|---|---|---|
| | | MTAP$^{def}$ | MTAP$^{prof}$ | MTAP$^{def}$ | |
| Patient Characteristics | | N (%) | N (%) | N (%) | N (%) |
| All | | 4 (100%) | 10 (100%) | 7 (100%) | 21 (100%) |
| Age - median (min, max) | | 65 (57, 80) | 69 (49, 79) | 71 (68, 80) | 70 (49, 80) |
| Gender | F | 2 (50%) | 2 (20%) | 4 (57%) | 8 (38%) |
| | M | 2 (50%) | 8 (80%) | 3 (43%) | 13 (62%) |
| Race | Asian | 1 (25%) | 3 (30%) | 0 (0%) | 4 (19%) |
| | Black | 0 (0%) | 0 (0%) | 3 (43%) | 3 (14%) |
| | Latino/Hispanic | 1 (25%) | 0 (0%) | 0 (0%) | 1 (5%) |
| | White | 2 (50%) | 7 (70%) | 4 (57%) | 13 (62%) |
| ECOG PS | 0 | 1 (25%) | 6 (60%) | 3 (43%) | 10 (48%) |
| | 1 | 2 (50%) | 3 (30%) | 4 (57%) | 9 (43%) |
| | 2 | 1 (25%) | 1 (10%) | 0 (0%) | 2 (9%) |
| Sites of metastasis | Nodal only | 1 (25%) | 6 (60%) | 3 (43%) | 10 (48%) |
| | Visceral (lung, bone, liver, or others) | 3 (75%) | 4 (40%) | 4 (57%) | 11 (52%) |
| Prior Therapy | Gem/Cis | 2 (50%) | 7 (70%) | 3 (43%) | 12 (57%) |
| | Immune checkpoint therapy | 0 (0%) | 1 (10%) | 5 (71%) | 6 (28%) |
| | GTA | 0 (0%) | 2 (20%) | 2 (29%) | 4 (19%) |
| | MVAC$^a$ | 0 (0%) | 5 (50%) | 1 (14%) | 6 (28%) |
| | CGI | 1 (25%) | 1 (10%) | 1 (14%) | 3 (14%) |
| | IAGEM | 0 (0%) | 1 (10%) | 0 (0%) | 1 (5%) |
| | Other Prior Therapy$^b$ | 1 (25%) | 2 (20%) | 1 (14%) | 4 (19%) |
| Number of Prior Regimens | 1 | 4 (100%) | 4 (40%) | 3 (43%) | 11 (53%) |
| | 2 | 0 (0%) | 3 (30%) | 1 (14%) | 4 (19%) |
| | 3 | 0 (0%) | 3 (30%) | 3 (43%) | 6 (28%) |

$^a$One MTAP$^{prof}$ retrospective patient with MVAC as metastatic therapy was included. All other MVAC therapy was adjuvant or neo-adjuvant therapy.
$^b$Other prior therapies include Gemcitabine and Cyclophosphamide ($n = 1$), carboplatin and paclitaxel ($n = 2$), and paclitaxel ($n = 1$).
*MTAP$^{def}$* MTAP deficient, *MTAP$^{prof}$* MTAP proficient, *ECOG PS* Eastern Cooperative Oncology Group performance status, *Gem/cis* gemcitabine/cisplatin, *GTA* gemcitabine, taxotere, and adriamycin, *MVAC* methotrexate, vinblastine, adriamycin, and cisplatin, *CGI* cisplatin, gemcitabine, and ifosfamide, *IAGEM* ifosfamide, adriamycin, and gemcitabine.

these findings, a much higher proportion of MTAP$^{def}$ UC cells underwent apoptosis in response to pemetrexed treatment as compared to MTAP$^{prof}$ UC cells, as measured by proportion of cells in sub-G1 phase (Fig. 2c) and poly (ADP-ribose) polymerase-1 (PARP1) cleavage (Supplementary Fig. 2c). Of note, these striking differences in apoptosis were observed even though MTAP$^{def}$ UC cells were treated with pemetrexed at a 40-fold lower concentration (0.5 µM) as compared to MTAP$^{prof}$ UC cells (20 µM) (Fig. 2c), suggesting a wide therapeutic window of pemetrexed for MTAP$^{def}$ UC.

Next, we assessed whether pemetrexed treatment led to increased DNA double-strand breaks (DSBs) in correlation with MTAP$^{def}$. DSBs were evaluated via the detection of phosphorylated histone H2A, X variant (γH2AX) and foci of p53-binding protein 1 (53BP1). The 4 MTAP$^{def}$ cell lines had significantly higher γH2AX and 53BP1 signals as compared to the 4 MTAP$^{prof}$ cell lines (Fig. 3a–d). As a positive control, the γH2AX response to treatment with gemcitabine was not different between MTAP$^{def}$ and MTAP$^{prof}$ cell lines (Supplementary Fig. 4b, c). Furthermore, we assessed the in vitro distortion of nucleotide monophosphate (NMPs) pools upon exposure to pemetrexed. At baseline, we observed that MTAP$^{def}$ cell lines had a trend for lower levels of AMP, CMP, UMP and GMP as compared to MTAP$^{prof}$ cell lines. However, the difference was not significant. Upon treatment with pemetrexed (5 um), NMPs increased by several folds. The fold rise of NMPs had a higher trend in MTAP$^{def}$ as compared to MTAP$^{prof}$ cell lines. However, the difference was not significant between the MTAP$^{def}$ and MTAP$^{prof}$ cell lines (Supplementary Fig. 5a, b).

Additionally, we analyzed an independent large-scale cancer cell line drug sensitivity database, the Genomics of Drug Sensitivity in Cancer (GDSC)[31], We found that MTAP$^{def}$ UC cell lines from the GDSC database ($n = 5$) were significantly more susceptible to methotrexate with a mean IC$_{50}$ of 0.25 µM as compared to a mean IC$_{50}$ of 1.3 µM for the MTAP$^{prof}$ cell lines ($n = 6$) ($P = 0.036$) (Supplementary Fig. 6a).

To further confirm our hypothesis that tumor MTAP deficiency leads to antifolate sensitivity, we carried out in vivo experiments in xenograft models of UC, which demonstrated that the MTAP$^{prof}$ HT-1376 tumor was resistant to pemetrexed (Fig. 2d) whereas the MTAP$^{def}$ UM-UC-3 tumor was sensitive to pemetrexed (Fig. 2e). As expected, in vivo pemetrexed treatment also resulted in significantly higher levels of DNA damage (as shown by 53BP1 staining) in MTAP$^{def}$ UM-UC-3 tumors as compared to the MTAP$^{prof}$ HT-1376 tumors (Supplementary Fig. 7a–f).

**Pemetrexed is effective against MTAP knockdown UC cells.** To verify that pemetrexed sensitivity is attributed to loss of MTAP instead of other co-deleted genes in the 9p21.3 region[32,33], we selectively knocked down the *MTAP* gene (MTAP$^{KD}$) in HT-1376 (a cell line with intact P16 expression), which led to substantial loss of MTAP protein expression and significant accumulation of its substrate MTA in the culture medium (Fig. 2f) in two different colonies: shMTAP2 and shMTAP3. Viability of the MTAP$^{KD}$ HT-1376 cell lines (shMTAP2 and shMTAP3) in response to pemetrexed treatment was much lower than parental and scramble control shRNA (Fig. 2g). Increased sensitivity to pemetrexed with MTAP$^{KD}$ was also reproducible using a mixture of shRNA to knock down the MTAP gene in HT-1376 (Supplementary Fig. 2d, e).

**L-alanosine, inhibitor of de novo adenine synthesis, is effective against human MTAP$^{def}$ UC cell lines.** To further verify that the

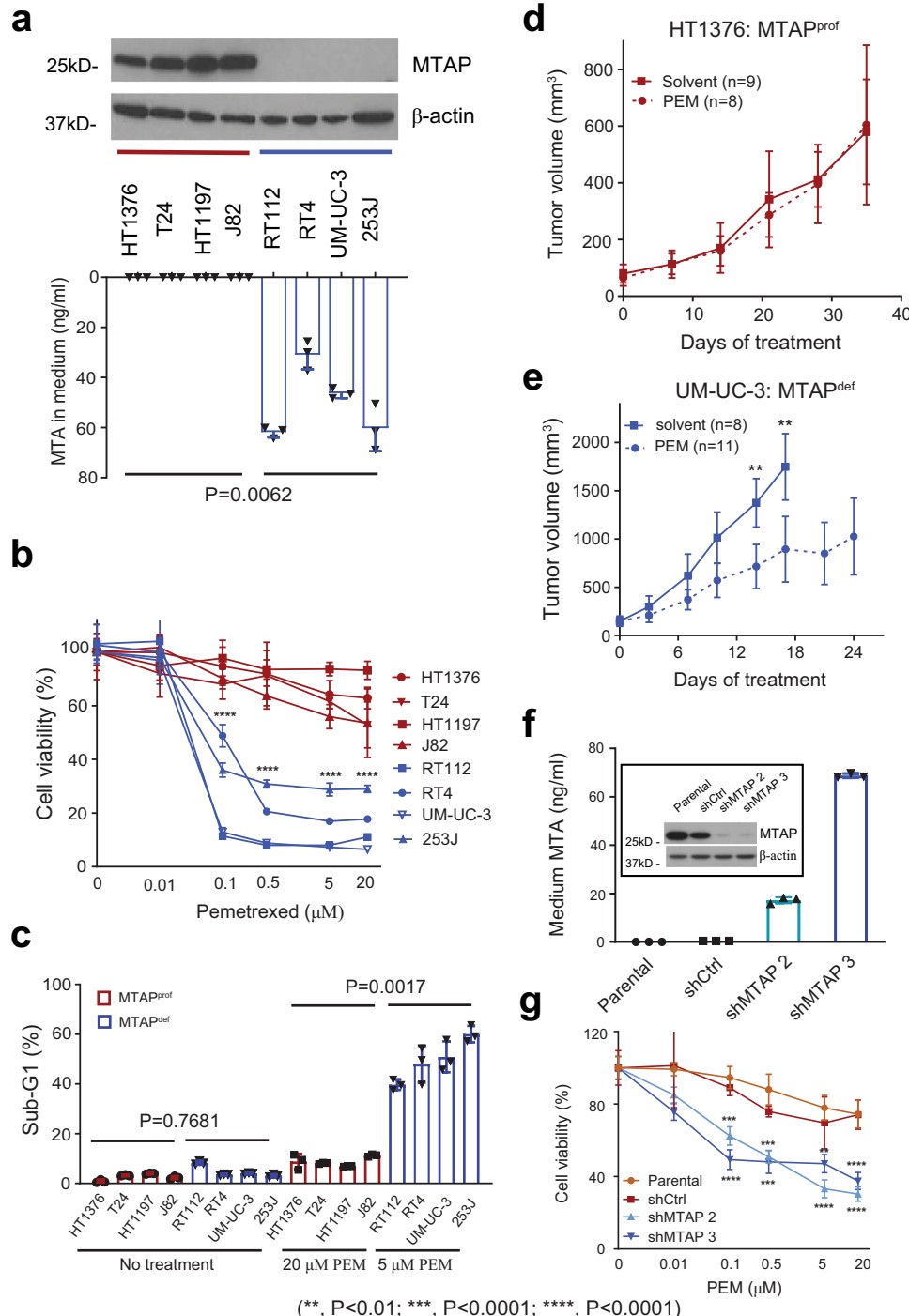

**Fig. 2 In vitro and in vivo effects of pemetrexed on human UC based on MTAP protein status. a** Western blot verification of MTAP protein loss in four MTAP$^{def}$ vs four MTAP$^{prof}$ human UC cell lines paired with UHPLC-ESI/triple quadrupole mass spectrometry measuring MTA concentration in cell media of human UC cell lines (ng/mL). MTA levels are significantly higher in MTAP$^{def}$ compared to MTAP$^{prof}$ cell lines $P$ value was calculated by Welch $T$ test. **b** Sub-G1 analysis of MTAP$^{def}$ and MTAP$^{prof}$ cell lines when treated with 0.5 μM and 20 μM of pemetrexed (PEM). $P$ value was calculated by Mann-Whitney test. **c** Cell viability after treatment with increasing concentrations of pemetrexed is significantly higher in MTAP$^{prof}$ compared to MTAP$^{def}$ cell lines $P$ value was calculated by two-way ANOVA. **d, e** Xenograft tumor volume in HT-1376 and UM-UC-3 when treated with pemetrexed vs solvent. $P$ value was calculated by two-way ANOVA. **f** Western blot and UHPLC-ESI/triple quadrupole mass spectrometry confirmation of *MTAP* knockdown in two HT-1376 cell lines: shMTAP2 and shMTAP3. **g** Pemetrexed resulted in significantly lower viability in the transfected HT1376 compared to the parental cell lines, with or without transduction of the lentivirus control vector. $P$ value was calculated by two-way ANOVA. Data in a (bottom panel), **b**–**e**, f (bottom panel), and **g** are presented as mean $+/-$ SD.

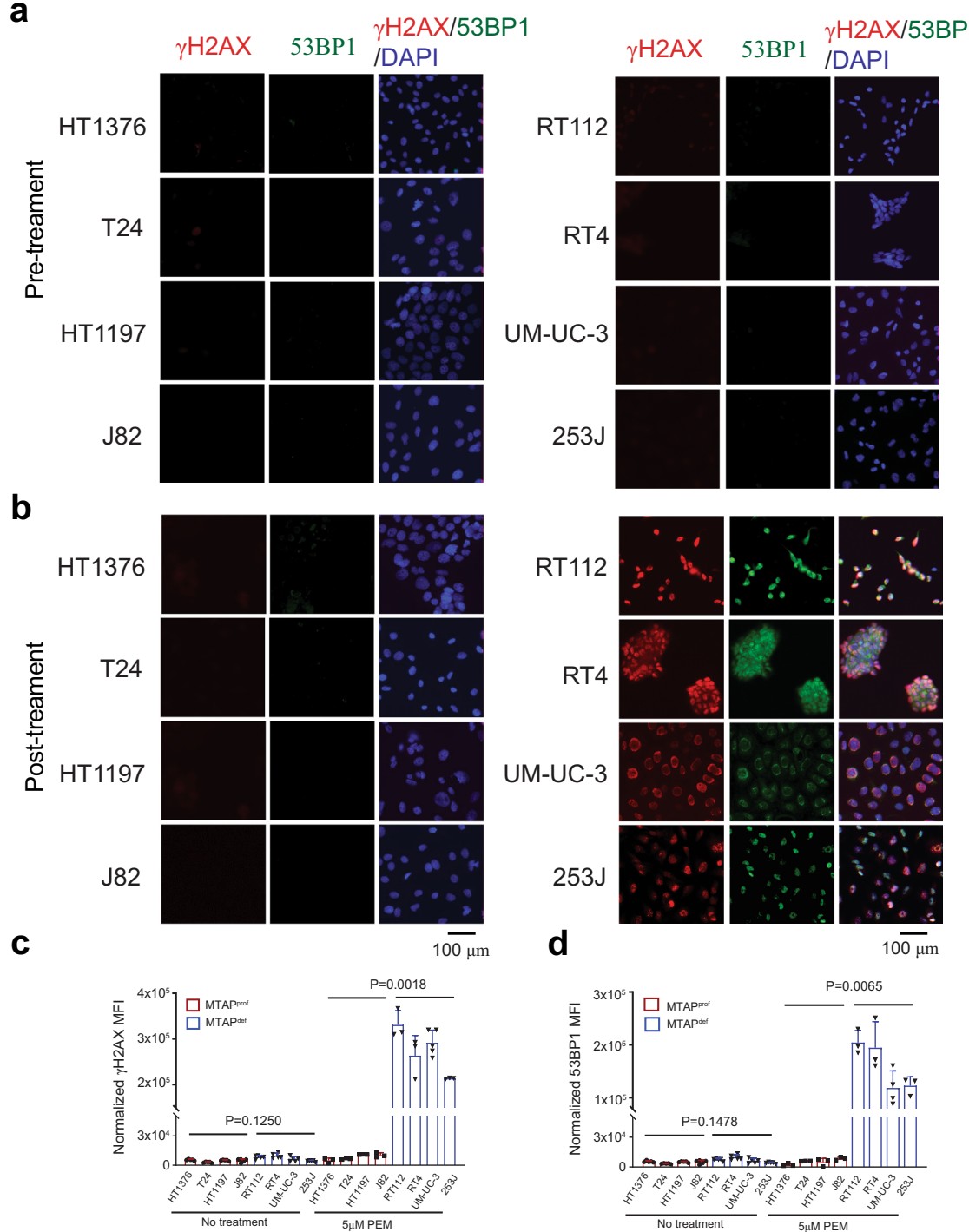

**Fig. 3 Pemetrexed (PEM) induces significantly higher DNA damage in MTAP^def cell lines as compared to MTAP^prof cell lines.** Eight human bladder cancer cell lines were treated with no treatment (**a**) or 5 μM of PEM (**b**) for 24 h. Cells were then fixed and stained with γ-H2AX (red), and 53BP1 (green) antibodies as well as DAPI stain (blue). Images were quantified with ImageJ for γ-H2AX (**c**) and 53BP1 (**d**). Data represent mean ± SEM of four fields and analyzed with Welch *T* test.

synthetic lethal interaction between inhibition of de novo adenine synthesis and tumor MTAP deficiency is a pathway-specific interaction, we tested the sensitivity of MTAP^def and MTAP^prof cell lines to L-alanosine, a well-known amino acid analog and a potent inhibitor of de novo adenosine monophosphate (AMP) synthesis. Our data indicate that MTAP^def cell lines were significantly more susceptible to cytotoxicity from L-alanosine treatment as compared to MTAP^prof cell lines (Supplementary Fig. 8a).

**Pemetrexed-containing chemotherapy is clinically effective against CDKN2A^lo/MTAP^lo lung adenocarcinoma.** To further validate the correlation between tumor MTAP loss and sensitivity to antifolate agents, we took advantage of the BATTLE-2 trial[34] that enrolled patients with metastatic nonsmall cell lung cancer, some of which were treated with pemetrexed-containing chemotherapy. Among a total of 200 patients, we identified a cohort of 72 patients with lung adenocarcinoma (LUAD) who were

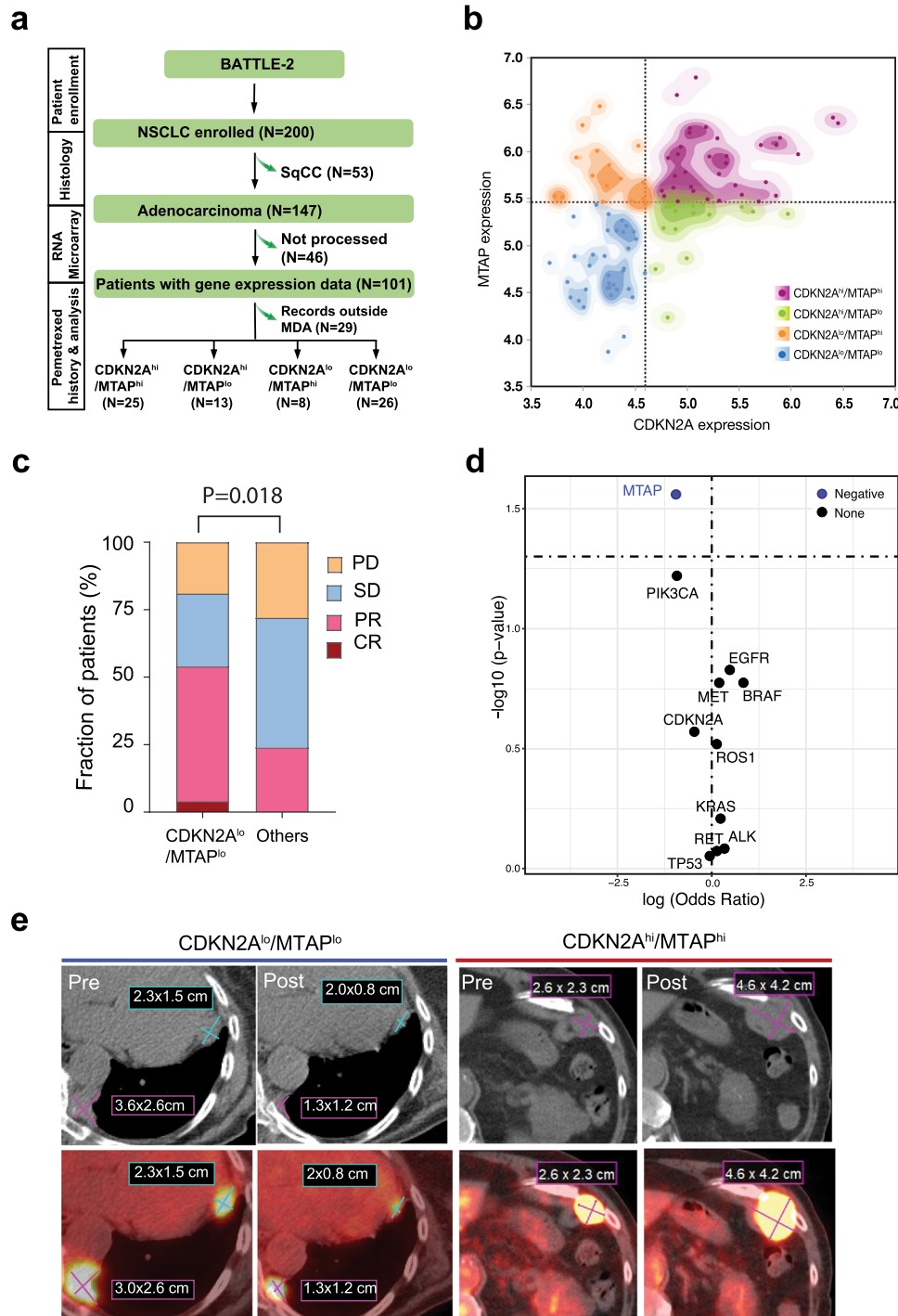

**Fig. 4 MTAP deficiency leads increased sensitivity to folate-based therapy in lung adenocarcinoma. a** Retrospective analysis schema for the BATTLE-2 trial. **b** Scatterplot of CDKN2A and MTAP RNA expression divided into four distinct groups with CDKN2A$^{lo}$/MTAP$^{lo}$ and CDKN2A$^{hi}$/MTAP$^{hi}$ having no overlap. MTAP cutoff value was 5.44 and CDKN2A cutoff value was 4.6 **c** Response rates to pemetrexed-based therapy in CDKN2A$^{lo}$/MTAP$^{lo}$ vs all other groups. Difference is statistically significant by two-sided Fisher's exact test ($p = 0.0115$). **d** Generalized linear regression model evaluating the correlation of 10 most altered genes in lung cancer beside MTAP to estimate the odds ratio and $p$ value for each gene independently. Genes with an odds ratio >1 (log (odds ratio) >0) and a $p$ value <0.05 are considered to be positively correlated with response. Genes with an odds ratio <1 (log (odds ratio) <0) and a $p$ value <0.05 are considered to be negatively correlated with response. Adjustments were made for multiple gene comparisons and q value are presented in supplementary table S5. **e** Example of a patient with metastatic CDKN2A$^{lo}$/MTAP$^{lo}$ lung adenocarcinoma who partially responded to pemetrexed-based therapy compared to a patient with CDKN2A$^{hi}$/MTAP$^{hi}$ lung adenocarcimoma who progressed after pemetrexed-based therapy.

treated with pemetrexed plus carboplatin and had tumor RNA expression data on both the MTAP and CDKN2A genes (Fig. 4a). Based on the co-expression of *MTAP* and *CDKN2A*, patients were divided into four groups: CDKN2A$^{hi}$/MTAP$^{hi}$, CDKN2A$^{hi}$/ MTAP$^{lo}$, CDKN2A$^{lo}$/MTAP$^{hi}$, and CDKN2A$^{lo}$/MTAP$^{lo}$ (Fig. 4b). Given the limitation of having only RNA expression data without copy number data, we chose CDKN2A$^{lo}$/MTAP$^{lo}$ as a representative of 9p21 loss and compared it to all other groups

| Patient Characteristics | | MTAP$^{lo}$/CDKN2A$^{lo}$ | | Others | |
|---|---|---|---|---|---|
| | | N$^a$ | (%) | N$^a$ | (%) |
| **All** | | **26** | **(100%)** | **46** | **(100%)** |
| Age – median (min, max) | | 60.5 | (34.0, 82.0) | 59.0 | (26.0, 76.0) |
| Gender | Female | 16 | (67%) | 23 | (51%) |
| | Male | 8 | (33%) | 22 | (49%) |
| Race | Asian | 1 | (4%) | 1 | (2%) |
| | Black | 1 | (4%) | 2 | (4%) |
| | White | 22 | (92%) | 42 | (93%) |
| ECOG PS | 0 | 1 | (4%) | 2 | (4%) |
| | 1 | 18 | (75%) | 40 | (89%) |
| | 2 | 5 | (21%) | 3 | (7%) |
| Kras Mutation | No | 17 | (71%) | 31 | (69%) |
| | Yes | 7 | (29%) | 14 | (31%) |
| EGFR mutation | No | 14 | (58%) | 37 | (84%) |
| | Yes | 10 | (42%) | 7 | (16%) |
| Smoking Status | Current | 2 | (8%) | 6 | (13%) |
| | Former | 10 | (42%) | 20 | (44%) |
| | Never | 12 | (50%) | 19 | (42%) |
| Line of Therapy | 1 | 16 | (62%) | 32 | (70%) |
| | 2 | 8 | (31%) | 8 | (17%) |
| | 3 | 1 | (4%) | 5 | (11%) |
| | 4 | 1 | (4%) | 0 | (0%) |
| | 6 | 0 | (0%) | 1 | (2%) |

**Table 2 Baseline patient characteristics of patients with lung adenocarcinoma at the start of pemetrexed treatment.**

*MTAP$^{lo}$* MTAP below-median expression, *CDKN2A$^{lo}$* CDKN2A below-median expression, *ECOG PS* Eastern Cooperative Oncology Group performance status.
$^a$Patients with unavailable information for a specific feature were not included, so counts may not always sum to 26 and 46.

(presumably without 9p21 loss). We noted that objective responses (54%) were significantly higher in CDKN2A$^{lo}$/MTAP$^{lo}$ patients compared with that (24%) in the rest of patients ($p = 0.018$ by two-sided Fisher's exact test) (Fig. 4c and Supplementary Table 4). Of note, among the baseline characteristics of this cohort of patients with LUAD (Table 2), we observed that epidermal growth factor receptor (*EGFR*) alterations were more prevalent among *CDKN2A$^{lo}$/MTAP$^{lo}$* patients as compared to the rest (42% vs. 16%, $P = 0.04$). To assess the impact of frequently altered genes in lung cancer, we performed a generalized linear model. However, none of the assessed genes showed a significant positive nor negative association with response to therapy except *MTAP* (Fig. 4d, Supplementary Table 5). An example of positron emission tomography-computerized tomography of a responding patient with a *CDKN2A$^{lo}$/MTAP$^{lo}$* tumor and a non-responding patient with a *CDKN2A$^{hi}$/MTAP$^{hi}$* tumor is shown in Fig. 4e.

## Discussion

Immunotherapy has revolutionized the therapeutic landscape across different cancer types including UC[35]. However, in the foreseen future, chemotherapy agents either alone or in combination with immunotherapy will continue to be one of the main therapeutic modalities for most cancers. In the era of precision medicine, predictive biomarkers, such as genomic alterations for targeted therapy[36] and PD-L1 expression and tumor mutation burden for immunotherapy[8,37], are playing an increasingly important role to guide selection of therapeutic strategies. However, there are currently no reliable biomarkers other than tumor histology to guide selection of chemotherapy regimens. In this study, we evaluated the clinical outcomes of patients with UC who were treated with pemetrexed stratified by tumor MTAP protein expression as a surrogate marker for 9p21 status. The

sensitivity to antifolates created by MTAP loss was further supported by in vitro and in vivo experiments using human UC cell lines. This sensitivity seemed to be unique to MTAP loss rather than CDKN2A loss, based upon the data that MTAP knockdown in the MTAP$^{prof}$ cell line HT-1376 rendered it sensitive to pemetrexed cytotoxicity whereas loss of the CDKN2A gene product p16 in MTAP$^{prof}$ cell line T24 did not result in sensitivity to pemetrexed. Mechanistically, antifolates are known to disrupt nucleotide biosynthesis and DNA replication, leading to DNA breakage and programmed cell death[38,39]. Our findings support this mechanism in MTAP$^{def}$ and a future direction of ours is to capitalize on this pathway and test combinations that target folates and DNA repair.

Furthermore, in an independent cohort of patients with LUAD, where pemetrexed is commonly used in frontline platinum-based chemotherapy combinations, we found that CDKN2A$^{lo}$/MTAP$^{lo}$ (surrogate marker for 9p21 loss) patients had significantly higher clinical response rates to such therapy regimens as compared to patients with likely 9p21 intact tumors. Collectively, our data support the hypothesis that tumor MTAP deficiency creates a metabolic vulnerability for therapy with antifolate agents such as pemetrexed. To optimize the clinical utility of our findings, we aim to investigate MTAP among a combination of biomarkers that have been associated with responsiveness to pemetrexed such as TS and DHFR[40].

We report an observation from clinical data, which we confirmed by performing mechanistic studies and our investigation on MTAP deficiency and antifolate sensitivity is, to our best knowledge, the first in urothelial cancer. One of the few common features among the 21 analyzed patients was their receipt of pemetrexed as second-line therapy or beyond, although their exposure to prior lines of therapy varied. Despite these clinical heterogeneities, our study represents the first effort to assess clinical response to pemetrexed in correlation with a frequent tumor genomic alteration in UC. Our data demonstrate that clinical responses to pemetrexed are enriched in patients with UC containing MTAP loss. These data compare favorably to the previously published data on pemetrexed as a second-line therapy, which showed a modest RR of 28% (13 of 47 patients) in UC without biomarker correlation[41]. In addition, these clinical data in UC are supported by preclinical evidence using human UC cell lines and by an independent cohort of patients with LUAD. Given the association between low MTAP expression and *EGFR* alterations, it is plausible that antifolates and an EGFR-directed oral tyrosine kinase inhibitor (TKI) could demonstrate synergy. In fact, a recently published phase III randomized trial in patients with advanced LUAD harboring an *EGFR*-sensitizing mutation showed that adding pemetrexed and carboplatin chemotherapy to gefitinib significantly prolonged PFS and OS[42]. In UC, an association was reported between MTAP loss and *FGFR* alterations[43,44]. Therefore, our findings support testing the addition of antifolates to the FDA-approved fibroblast growth factor receptor (FGFR)-directed oral TKI in *FGFR*-mutant UC[36].

Pemetrexed has been reported to augment intra-tumor immune responses through increased T cell infiltration/activation along with modulation of innate immune pathways and as a result, enhanced the anti-tumor activity of anti-PD-L1 in pre-clinical models[45]. Based upon the immune-modulatory ability of pemetrexed and its pronounced cytotoxicity in MTAP$^{def}$ UC, it appears that pemetrexed can be combined with anti-PD-L1 to provide a highly effective treatment for patients with MTAP$^{def}$ UC. In fact, we have launched a phase II trial assessing a sequential combination of pemetrexed and avelumab (anti-PD-L1 antibody) in patients with MTAP$^{def}$ UC (NCT03744793). Although this trial led to an intentional, early termination of the pemetrexed monotherapy trial in MTAP$^{def}$ UC, we expect this tissue-rich,

sequential combination trial will shed light on the interaction between pemetrexed and ICT and provide superior therapeutic benefits to patients with MTAP[def] UC as part of 9p21 loss.

Although the term *synthetic lethality* classically refers to an interaction between two genetic events, synthetic lethality can also refer to cases in which the combination of a mutation and the action of a chemical compound causes lethality, whereas the mutation or compound alone is non-lethal[46]. Our data reveal a synthetic lethality strategy to exploit MTAP loss in UC, and potentially in LUAD, by inhibiting de novo adenine nucleotide synthesis, inducing DNA damage, and potentially distorting nucleotide pools using antifolate agents such as pemetrexed. Loss of *MTAP* along with adjacent genes such as *CDKN2A* in the chromosome region 9p21.3 represents one of the most frequent genomic defects existing in 14% of all malignancies and is associated with poor clinical outcomes[47–49]. Therefore, efforts to develop novel strategies to target MTAP[def] tumors have increased in recent years, resulting in the identification of a number of synthetic lethal inhibitors, including methionine adenosyl-transferase II alpha (MAT2A) and type I and type II protein arginine N-methyltransferase (PRMT) inhibitors[14,15,50,51]. In this report, we identify an opportunity for a synthetic lethal strategy using a safe agent[52], pemetrexed, in MTAP[def] tumors. These data may be extrapolated to other malignancies that harbor *MTAP* loss as part of chromosome region 9p21.3 deletion.

## Methods

### Urothelial carcinoma study population

*Trial design, patients, and treatment.* NCT02693717 is a single-arm non-randomized phase II clinical trial to evaluate pemetrexed disodium in previously treated metastatic MTAP[def] UC. MD Anderson Cancer Center (MDACC) and the National Cancer Institute (NCI) sponsored this trial. This trial was approved by the Institutional Review Boards (IRB) of MDACC (MDACC protocol 2015-0592). The primary objective of this trial was to evaluate the overall RR defined by Response Evaluation Criteria in Solid Tumors (RECIST) 1.1. The secondary objectives included: 1) evaluating PFS, 2) evaluating OS, 3) evaluating the safety of peme-trexed therapy, 4) collecting blood, urine, and tissue for future translational studies. All primary and secondary outcomes are reported in the manuscript. Enrollment was done between 9/1/2017 and 1/25/2019. Patients provided written consent and the signed consent form was scanned into the electronic medical record. Patients also received a copy of the signed consent form. Patients were not compensated for their participation. Patients were required to have histological confirmation of metastatic UC and sufficient tumor tissues for MTAP IHC testing. Patients who had received any non-antifolate-containing systemic therapy (including immu-notherapy) were eligible. The total estimated accrual for this trial was 25 patients. However, only seven patients were enrolled due to competing protocols and the opening of NCT03744793. Patients received pemetrexed disodium IV over 10 min on day 1. Courses repeated every 21 days in the absence of disease progression or unacceptable toxicity. Pre- and post-treatment blood and urine samples were collected for correlative biological analyses per an IRB-approved laboratory protocol MDACC PA17-0577.

*Historic cohort.* We retrospectively evaluated 14 metastatic UC patients treated with pemetrexed beyond first line at MDACC between 1/1/2014 and 7/1/2018 (MDACC PA17-0577). Four patients were MTAP[def] and 10 were MTAP[prof] upon IHC staining of tumor tissue. Details about this cohort are described in the "patients characteristics" paragraph in the Results section and in Table 1.

*Objectives and statistical plan.* We analyzed tumor responses to pemetrexed based on MTAP status. Patient characteristics were tabulated. The response was considered to be complete (CR) or partial (PR) per RECIST 1.1. All other responses, including not evaluated were considered non-responses. The proportion of responses for MTAP[def] vs. MTAP[prof] were compared with Fisher's exact test to accommodate the small number of patients. AEs graded according to the National Cancer Institute Common Terminology Criteria for Adverse Events (CTCAE) version 4.03. OS was defined as the time from the first day of pemetrexed treatment until death or last contact. PFS was defined as the time from the first day of pemetrexed treatment until disease progression, death, or last disease assessment. Patients alive and without progression at their last assessment were censored on their last date of assessment before starting a new treatment. Kaplan-Meier curves are presented for OS, and PFS. Only descriptive results are provided due to sample sizes being too small for comparison. Analyses were performed in SAS 9.4 (The SAS Institute Inc, Cary, NC). Kaplan-Meier curves were created in Stata 14.1 (StatCorp, College Station, TX).

### Lung adenocarcinoma study population

*Patients and treatment.* Among the 101 patients with LUAD enrolled in the BATTLE-2 study[34] and having enough tissue for gene expression analysis, we analyzed the distribution of CDKN2A and MTAP genes to separate patients with 9p21 deletion. Gene expression analysis was done by messenger RNA GeneChip Human Gene 1.0 ST Array from Affymetrix (Santa Clara, CA). CDKN2A and MTAP expression was divided into high (hi) or low (lo) at the median cutoff (Supplementary Fig. 5,A, B). Patients then were divided into four groups based on the co-expression of MTAP and CDKN2A: CDKN2A[hi]/MTAP[hi], CDKN2A[hi]/MTAP[lo], CDKN2A[lo]/MTAP[hi] and CDKN2A[lo]/MTAP[lo] (Fig. 3b).

*Objectives and statistical plan.* Next, we evaluated 72 patients with metastatic LUAD treated with pemetrexed-based chemotherapy that had available imaging for response assessment based on RECIST 1.1. The response was considered to be complete (CR) or partial (PR) per RECIST version 1.1. All other responses, including not evaluated were considered nonresponders. The proportion of responses for CDKN2A[lo]/MTAP[lo] vs others were compared with Fisher's exact test to accommodate the small number of patients. OS was defined as the time from the first day of pemetrexed treatment until death or last contact. PFS was defined as the time from the first day of pemetrexed treatment until disease progression, death, or last disease assessment. Patients alive and without progression at their last assessment were censored on their last date of assessment before starting a new treatment. Kaplan-Meier curves are presented for OS, and PFS.

### Generalized linear model

To evaluate the association of the 10 most altered genes in lung cancer with drug response, generalized linear model (GLM) was used to estimate the odds ratio and p-value for each gene in the lung cancer cohort ($n = 72$) independently, with 1 indicating a responder and 0 a non-responder. Genes with an odds ratio >1 (log (odds ratio) >0) and a $p$ value <0.05 are considered to be positively associated with responders. Genes with an odds ratio <1 (log (odds ratio) <0) and a $p$ value <0.05 are considered to be negatively associated with responders.

### Patient samples for tissue microarray IHC

Patients eligible for enrollment were those seen at Mayo Clinic (Scottsdale, Arizona) who were >18 years old, able to provide informed consent, and undergoing evaluation for genitourinary diseases. Patients were contacted during routine clinical visits or in preoperative settings within Mayo Clinic departments and divisions, including urology, radiation oncology, pathology, and medical oncology. Patients were excluded if they declined to participate or if the banking of their biospecimens would compromise the availability of tissue for diagnosis and standard clinical care. The protocol for collecting biospecimens, the process for consenting patients, and the current informed consent form were approved by the Mayo Clinic IRB (protocol no. 08-000980). Patients were enrolled from June 1, 2010, through January 1, 2013. Patients provided written consent and the signed consent form was scanned into the electronic medical record. Patients also received a copy of the signed consent form.

### Genomics of drug sensitivity in cancer (GDSC) data

We analyzed an independent large-scale cancer cell line drug sensitivity dataset, the GDSC[31]. Analysis was performed using IC$_{50}$ values (drug concentration that reduces viability by 50%). MTAP deficiency in cell lines was determined from copy number variation (CNV) in MTAP from the Cancer Cell Line Encyclopedia (CCLE)[53]. Log2 ratio cutoff values $<-3$ were used to detect deep deletions for each CCLE pan-cancer cell line. Only cell lines with CNV status consistent with low expression of MTAP were maintained. UM-UC3 was excluded as the IC$_{50}$ value in the public database was not reproducible with our repeated experiments using methotrexate. 5637 was excluded as it contained an amplification of the *MTAP* gene. However, the MTAP protein was only expressed in the nucleus based upon our IHC study. Therefore, the enzymatic activity of MTAP was likely restricted and functionally represented an MTAP[def] cell line. We tested the difference in IC$_{50}$ between UC cell lines with or without deep deletion in MTAP by using a one-sided Wilcoxon rank-sum test.

### Mice, cell lines, and reagents

Athymic nude 6-week-old mice were purchased from the Charles River Laboratory. All in vivo experiments used male mice. All cell lines were originally obtained from ATCC (Manassas, VA), except for RT112 from Creative Bioarray (Shirley, NY). HT-1376, HT-1197, J82, and UM-UC-3 were maintained in MEM supplemented with 10% FBS, 2 mM L-glutamine, and 0.1 mM non-essential amino acids. 253 J was maintained in DMEM plus 10% FBS. RT112 was maintained in RPMI1640 plus 10%FBS. T24 and RT4 were grown in McCoy's 5 A plus 10% FBS. Only mycoplasma-free cultures were used. Pemetrexed (Alimta®) was obtained from Eli Lilly & Co (Indianapolis, IN). L-alanosine was purchased from MedKoo Biosciences (Morrisville, NC). Gemcitabine was obtained from SellekChem (Houston, TX). Adenine and 5′-deoxy-5′-methylthioadenosine (MTA) were obtained from MilliporeSigma (Burlington, MA).

### Antibodies and Western blot

For Western blotting, cells were homogenized in RIPA buffer (Cell Signaling, Danvers, MA) plus phosphatase and protease inhibitor

cocktail, 10 mM NaF, 1 mM PMSF and 1 mM Na3VO4. Protein extracts were purified, and concentrations were measured with Pierce Protein Assay kit (Thermo Fisher Scientific). The following antibodies were used in this study: Anti-MTAP antibody (Rabbit polyclonal, 1:2000) from ProteinTech, anti-poly (ADP-ribose) polymerase-1 (PARP-1) antibody (Rabbit monoclonal, 1:2000) from Cell Signaling (Danvers, MA), and β-actin (Mouse monoclonal, 1:10000) from MilliporeSigma (Burlington, MA). We assessed the cleavage of PARP-1 using Western blot. PARP-1 cleavage by caspase-3 and −7 during apoptosis into a 24 kDa N-terminal fragment and an 89 kDa fragment has been used as a marker of apoptotic animal cells[54]. Goat anti-rabbit and goat anti-mouse secondary antibodies were obtained from Santa Cruz Biotechnology (Dallas, TX). The chemiluminescence signals were developed with Bio-Rad ChemiDoc XRS + System (Bio-Rad, Hercules, CA).

**Immunohistochemistry and immunofluorescent staining**. MTAP immunohistochemical staining was performed on a 4 μm unstained slide of a formalin-fixed paraffin-embedded (FFPE) tissue microarray using an automated immunohistochemical stainer (Leica Bond III, Leica Biosystems, Buffalo Grove IL). Following deparaffinization and rehydration, antigen retrieval was performed at 95 °C for 30 min with citrate buffer, pH 6.0. Endogenous peroxidase was blocked with 3% peroxide for 5 min. Primary anti-methylthioadenosine phosphorylase antibody (ProteinTech, Rosemont, IL) was applied at 1:1200 dilution for 15 min. Primary antibody detection was carried out using a commercial polymer system (Bond Polymer Refine Detection, Leica Biosystems, Buffalo Grove IL), and stain development was achieved by incubation with DAB and DAB Enhancer (Leica Biosystems, Buffalo Grove IL). Tumor cells were assessed for the presence or absence of labeling, defined as the presence of any staining or complete loss of staining, respectively.

γ-H2AX and 53BP1 immunofluorescent staining for 8 cell lines was performed with Nunc Lab-Tek II 8-well chamber slides system (ThermoFisher, Waltham, MA). Briefly, cells were plated in individual wells at about 70–80% confluence. Five μM pemetrexed or 5 μM gemcitabine was then added into medium, and cells were cultured for additional 24 h. At the end of treatment, anti-γ-H2AX (1:200, MilliporeSigma) or anti-53BP1 (1:200, Cell Signaling) were added to slides after fixation and permeabilization. For 53BP1 staining on FFPE sides, deparaffinization, rehydration, and antigen retrieval were processed the same as immunohistochemistry. Antibody (1:200) was then placed to tissues and incubated for 30 min at room temperature. Positive signals were captured with EVOS M5000 fluorescent microscope (ThermoFisher, Waltham, MA). At least 3 areas at 20x magnification were taken from each slide for further data analysis. Fluorescence processing and quantification were performed with ImageJ[55].

**Cell viability assay**. Cell viability was determined by MTT assay. Briefly, cells were plated at a density of 2,000 cells per well in 96-well cell culture plates. Cells were then treated with PEM or L-alanosine at various doses for 72 h. Each dose was tested at least in triplicate. At the end of treatment, 0.5 mg/mL MTT was added into the cultures for 2 h. Insoluble MTT metabolites were dissolved with DMSO and absorbance was measured at 530 nm with background subtraction. IC$_{50}$ values were then calculated with Graphpad Prism 7.0 software.

**MTA and nucleotide pool quantification using ultra-high-performance liquid chromatography-electrospray ionization (UHPLC-ESI)-triple quadrupole mass spectrometry**. For MTA measurement, 10$^6$ cells were plated in 10-cm dishes in triplicate and cultured for 48 h. Culture media and cell pellets were collected separately and kept at −80 °C until measurement. At the day of quantitation, cell pellet was vortexed with 0.6 mL of methanol containing 25 ng of MTA/adenosine internal standard, centrifuged at 12,000 rpm (15300 × g) for 5 min. The supernatant was decanted and evaporated under nitrogen. Each sample was reconstituted in 100 μL water. Cell line media (250 μL) was processed as cell pellets. An Agilent UHPLC Infinity II system was combined with a Chromolith reverse phase column (100 × 2 mm, 1.5 μm, MilliporeSigma, Billerica, MA). Mobile phases used were (A) 0.1% formic acid and water and (B) 0.1% formic acid and methanol at a flow rate of 0.3 mL/min. The isocratic elution was held at 20% A for 2 min, and the gradient was ramped up to 95% B in 5 min then held at 95% B for 2 min before being reduced to 80% A and then re-equilibrated for 2 mins. The column temperature was 30 °C and the injection volume was 5.0 μL. Molecules were introduced through a Jet Stream electrospray ionization source to an Agilent 6495 triple quadrupole mass spectrometer (Agilent, Santa Clara, CA). The source was operated in a positive ion mode under optimum conditions. Nucleosides were identified by retention time and parent product ion transitions for MTA quantitative ion (m/z 297.81–36.1) and qualitative ion (m/z 297.81–119.0), and quantified using MassHunter software (Agilent, Santa Clara, CA).

A negative ionization method was developed on the Agilent 6495 LC-MS-MS to determine mono and tri-phosphate nucleotide levels. A Phenomenex SEPAX column (2.1 × 50 mm, 3.5 μm) was used with a gradient of Buffer A: 5 mM ammonium acetate and Buffer B: Acetonitrile. The cell pellets were harvested and extracted using 150 ul methanol/water (85/15) spiked with 25 ng/mL C13-CTP, C13-ATP, and C13-dGTP, the vortexed for 1 min and placed on ice for 10 min. After centrifugation at 11,000 rpm (12,800 × g) at 4 °C for 15 min, the supernatant

was diluted with 1350 μl 60% Methanol before loading onto an OASIS HLB cartridge (Waters Corp., Milford, MA) that had previously been activated with methanol and water according to the SPE protocol. The final eluent (200 ul) was derivatized with MBSTFA (75 μl) and the derivatization reaction was completed over 5 min with consistent vortex. The samples were then centrifuged at 11,000 rpm for 15 min and 10 μl of the supernatant was injected in triplicate. Under optimal conditions, the lower limit of quantitation (LLOQ) was 0.1 ng/mL[56].

**MTAP knockdown**. Lentiviral particles of human short hairpin RNAs (shRNAs) specific to the *MTAP* gene and control scramble shRNA were purchased from Santa Cruz Biotechnology (Dallas, TX). Transduction of the HT-1376 cell line was carried out according to the manufacturer's instructions. Briefly, cells were cultured in 12-well plates to reach 50% confluence, transduced with lentiviral particles, split into 100 mm plates, and then selected with 2 μg/mL puromycin-containing culture medium. Single-cell colonies were selected and expanded in puromycin-containing culture medium. RT-PCR and Western blot were then used to confirm the extent of individual MTAP gene knockdown.

**Determination of cell cycle distribution by flow cytometry**. Cells were plated in 6-well plates and treated with pemetrexed for 72 h. At the end of treatment, cell cycle distribution was determined by propidium iodide (PI) staining (MilliporeSigma, Burlington, MA). Cells were collected and adjusted to a concentration of 10$^5$ cells/mL, fixed by 75% ethanol for more than 1 h, and then treated with RNase A for 30 min at room temperature. After 1 h incubation with PI staining buffer at room temperature, FACSCanto II flow cytometry (BD Biosciences, San Jose, CA) was used to detect cell cycle[57]. When apoptotic cells are stained with PI and analyzed with a flow cytometer, they display a broad G1 (hypodiploid) peak, which can be easily discriminated from the narrow peak of cells with normal (diploid) DNA content in the red fluorescence channels. Each experiment was repeated at least three times.

**Xenograft animal models and in vivo toxicity studies**. Animal experiments were carried out under conditions adhering to approved protocols from the Institutional Animal Care and Use Committee at MDACC. Briefly, anesthetized mice were injected in the right flank subcutaneously with 106 UM-UC-3, HT-1376, HT-1376/shCtrl, or HT-1376/shMTAP cells at day 0. Tumor growth was monitored by measuring tumor size with a caliper twice a week, and pemetrexed treatment was initiated once tumor size reached about 50–100 mm$^3$. The dose of pemetrexed for both models was 200 mg/kg with a frequency of three times a week. Tumor volume was calculated using a caliber. Mice were euthanized when tumor size reached 1.5 cm in diameter, ulceration reached 0.2 cm, or moribund occurred, and were recorded as deaths.

**Statistical analysis of in vitro and in vivo data**. All measurement data were expressed as mean ± standard deviation. If not stated otherwise, comparisons between groups were made using the *t* test. *P* values less than 0.05 were considered significant.

**Reporting summary**. Further information on research design is available in the Nature Research Reporting Summary linked to this article.

## Data availability

The data generated or analyzed during this study are included in this published article, its Supplementary Information and Source Data files. For TCGA BLCA cohort shown in Fig. 1b, the genomic data can be retrieved from NCI Genomic Data Commons (NCI-GDC: https://gdc.cancer.gov). Publicly available datasets pertaining to Supplementary Fig. 2a can be downloaded from https://depmap.org/portal/download/. Tissue microarray data pertaining to Fig. 1c, d is included in Source Data file. Clinical data pertaining Fig. 1e, g, h is included in Source Data file. Uncropped Western Blot gels and in vitro data pertaining to Fig. 2a–g is included in Source Data file. Clinical data and gene expression data pertaining to Fig. 4a–e is included in Source Data file. Additional data related to the current study, including the study protocol of the clinical trial NCT02693717, are available from the corresponding author (Jianjun Gao) on reasonable request that does not include confidential patient information. Source data are provided with this paper.

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

## Acknowledgements
We would like to acknowledge Gary Gallick, who sadly passed away, for his significant contributions to this research work. This study was supported in part by the Doris Duke Clinical Scientist Development Award (JJ, #2018097) granted to J.G. and L. W. In addition, this study was supported by the MD Anderson Physician Scientist Award, Khalifa Physician Scientist Award, Andrew Sabin Family Foundation Fellows Award, MD Anderson Faculty Scholar Award, David H. Koch Center for Applied Research of Genitourinary Cancers, Wendy and Leslie Irvin Barnhart Fund, Joan and Herb Kelleher Charitable Foundation, and NCI/NIH R01 CA254988-01A1 Award provided to J.G. This study was also supported by SMF Core Grant CA016672 (SMF) and the IRG start-up research funds provided to L.W. by MD Anderson Cancer Center (MDACC). We thank Dr. C. Liu, X. Liu, and P. Xiong from the MD Anderson Sequencing and ncRNA Program, Dr. E. J. Thompson and D. P. Pollock from the SMF Core for their excellent technical assistance. This work was in part supported by the generous philanthropic contributions to The University of Texas MD Anderson Lung Moon Shot Program and the MD Anderson Cancer Center Support Grant P30 CA01667. We would like to acknowledge the GEMINI Team at MD Anderson Cancer Center. P.M. is supported by a Young Investigator Award by the Kidney Cancer Association, a Career Development Award by the American Society of Clinical Oncology, and by the MD Anderson Khalifa Scholar Award. We thank all the patients who participated in this study. O.A. is supported by a Young Investigator Award by American Society of Clinical Oncology. J.A.R. has received grants from Varian Medical Systems. Figure 1f and Supplementary Fig. 1a were created with a licensed version of BioRender.com. The sponsor had no role in the study design, data collection and analysis or manuscript writing.

## Author contributions
J.G., L.W., and T.H. conceived and jointly supervised this study. O.A., Y.L., M.T.C., A.Y.S., L.L., and Q.W. collected samples and clinical information. J.C., Y.Z., W.W., J.S., S.R., A.G.H., X.Z., and M.T. carried out experiments. O.A, J.C., X.Y., G.H., C.C.G., P.M., C.J.L., T.H.H., J. Z., L.W., and J.G. analyzed and interpreted data. R.S.T. performed the statistical design and analysis of clinical data. E.E., M.Z., C.C.G., C. B., L.S., and B.C supervised the anatomical pathology staining. O.A., J.C., G.H., R.W., A.A-D., W.F.B., Jennifer Wang, J.M., A.S-R., V.P., J.Lewis, W.R., V.R., J.Lee, J.R., S.S, I.W., J.H., Jing Wang, and L.W. contributed to the generation of figures and tables. O.A., J.C., L.W., and J.G wrote and revised the manuscript. All authors approved the manuscript.

## Competing interests
Dr. Shah has honorarium with Pfizer, BMS, Exelixis and research funding from BMS, Eisai, and EMD Serono. Dr. Siefker-Radtke serves as a consultant for Janssen, Merck, the National Comprehensive Cancer Network, Lilly, Bristol-Myers Squibb, AstraZeneca, BioClin Therapeutics, Bavarian Nordic, Seattle Genetics, Nektar, Genentech, Inovio Pharmaceuticals, and EMD Serono. Dr. Siefker-Radtke has received research funding from the National Institute of Health, Michael and Sherry Sutton Fund for Urothelial Cancer, Janssen, Takeda, Bristol-Myers Squibb, BioClin Therapeutics, and Nektar. Dr. Campbell has served as a consultant or has provided non-branded educational lectures with honorarium with Pfizer, EMD Serono, AstraZeneca, Eisai, Apricity, Roche, Bristol Myers Squibb, and Merck. Dr. Gao serves as a consultant for ARMO Biosciences, AstraZeneca, Jounce, Nektar, and Pfizer. Dr. Msaouel has received honoraria for service on a Scientific Advisory Board for Mirati Therapeutics, Exelixis, and BMS, consulting for Axiom Healthcare Strategies, non-branded educational programs supported by Exelixis and Pfizer, and research funding for clinical trials from Takeda, BMS, Mirati Therapeutics, Gateway for Cancer Research, and UT MD Anderson Cancer Center. Jack A. Roth has consultancy, stock, Genprex, Inc.; patents issued and pending. Dr. Ho has received honoraria from Exelixis, Genentech, EMD-Serono, Pfizer, Macrogenics, Cardinal Health, Ipsen, and Aveo. The remaining authors declare no conflicts of interest.
