## [Peer Review File · Nature Communications]

MTAP deficiency creates an exploitable target for antifolate therapy in 9p21-loss cancersREVIEWER COMMENTS

Reviewer #1 (Remarks to the Author); expert on MTAP and folate metabolism:

In the study by Alhalabi et al, the authors report that Pemetrexed is more effective in MTAP deficient UC than in MTAP proficient UC. They find that this hold true in a panel of UC cell lines and that depleting MTAP from a MTAP proficient line led to increased sensitivity to pemetrexed. Similarly, they found that xenografts derived from a MTAP proficient cell line were relatively insensitive to pemetrexed as compared to xenografts derived from a cell line that is MTAP deficient. Lastly, in a lung cancer clinical trial, MTAP low patients had better responses to Carboplatin plus Pemetrexed than MTAP high patients. Although there are no data indicating whether the correlation with MTAP levels has something to do with either Carboplatin or Pemetrexed.

The data present are clear and convincing, for the most part, and as expected. Unfortunately, blockade of de novo purine biosynthesis in MTAP deficient tumors is not novel. Neither is specifically use of pemetrexed in MTAP deficient tumors. (see: Mol Cancer Ther 2011;10:495-504; Cancer Biol Ther 2011;11:627-632; J. Clin. Oncol. 37, 385–385 (2019); Sci Rep 10, 843 (2020)).

A possible way to increase the novelty of the work would be to assess whether the treatment with Pemetrexed leads to increased DNA damage, and/or distorted nucleotide pools in patient samples and if this correlates with MTAP levels. Can the authors determine, perhaps in the xenograft studies, if the MTAP deficient sensitivity to Pemetrexed has to do with accumulating MTA, or to inhibited de novo purine synthesis?

Reviewer #2 (Remarks to the Author); expert on clinical trial statistics:

NCOMMS-20-37775-T

MTAP deficiency creates an exploitable target for antifolate therapy in 9p21-loss cancers

Submitted to: Nature Communications
Report: 11/27/2020

According to the authors, gene MTAP deficiency is significantly associated with that of the tumor suppressor gene CDKN2A in the chromosome region 9p21; they reported that patients with MTAP UC deficient had higher treatment response rate (7/11=64%) than those with MTAP UC-proficient (1/10=10%). In another independent cohort study (n=72), they found tumor MTAP-deficient was significantly associated with response rate to pemetrexed-based chemotherapy. Numerical study results are presented to justify their finding. They hypothesize that tumor MTAP deficiency may provide metabolic vulnerability for the use of antifolate agents such as pemetrexed to effectively treat UC with 9p21 loss. Thus in study of the effects of treatment to 9p21-loss cancers, the authors propose to target the MTAP UC deficiency, and suggest this strategy to be tested in larger trials.

Major Comments:

The studies were based on MTAP-deficient and MTAP-proficient groups. I have the following comments from a biostatistician's point of view.

* The sample size in each group is pretty small (many of them < 10), as the authors themselves acknowledged "Our study has several limitations, including the limited sample size" (p.8, line 193). This makes the authors' claim very weak. For a claim to be valid, the sample sizes of each group should be at least 100 and with significant group mean difference (p-value < 0.05).

* The authors used the naive method (sample mean + - standard deviation, p.15, line 395-397) in their studies. This is OK if the treatment assignment (the selection of the MTAP deficient and proficient groups) is completely at random. In practice, often such assignments are not completely random, and then the naive method will give biased conclusions. In this case, the causal inference method, such as propensity score adjusted method or the doubly robust estimate method will give unbiased results. These methods are covariates adjusted and used extensively in modern clinical trials and the literature can be easily found on line. I suggest the authors also use causal inference methods to validate their results, this will make their conclusion much stronger.

* Often the treatment effect is affected by a group of genes, instead of a single gene such as MTAP. Although the effect of each single gene in that group is highly insignificant, but the collective group effect can be very significant. Such effect can be investigated using methods such as the rare-variants method (Morris and Zeggini, 2010; Yuan et al., 2012). The authors are encouraged to investigate this method also, possibly with collaboration of some statistician.

Minor comment

* If the authors' hypothesis/proposal on MTAP is the first, please emphasize it.

References

Morris AP and Zeggini E. (2010). An evaluation of statistical approaches to rare variant analysis in genetic association studies. *Genetic Epidemiology*, 34: 188-193.

Yuan, A., Chen, G., Zhou Y., Bentley, A., Rotimi, C. (2012). A novel approach for the simultaneous analysis of common and rare variants in complex traits, *Bioinformatics and Biology Insight*. 6: 1-9.

Reviewer #3 (Remarks to the Author); expert on urothelial clinical trials:

1. Patients benefiting from pemetrexed may include those beyond responders- e.g. durable stable disease ≥ 6 months. Including them in the benefiting category may help refine the analysis?
2. Authors could comment on any association with benefit from another antimetabolites-gemcitabine- which is frequently used in many cancers including urothelial and lung cancer.
3. Please discuss the clinical utility given that 10% of metastatic urothelial carcinoma and MTAP proficient tumors did respond to pemetrexed. A combination of markers may be more optimal?
4. Genomic analyses may not always report MTAP or 9p or p16 loss? Association of MTAP loss with TMB and other actionable activating mutations may be useful to evaluate and discuss (e.g. FGFR3 in urothelial and EGFR in lung). There is also a reported sensitivity of EGFR mutated lung cancer to pemetrexed?
5. Was there controlling for clinical prognostic factors in the clinical datasets, ie was there association of MTAP with pemetrexed response independent of major clinical prognostic factors, ie performance status, sites of metastasis?
6. Also, association of MTAP loss with clinical prognostic factors would be useful to report.

Re: NCOMMS-20-37775-T

Title: MTAP deficiency creates an exploitable target for antifolate therapy in 9p21-loss cancers

Dear Reviewers,

We would like to thank all of you for your insightful reviews and constructive comments on our manuscript. Your insights and queries have helped us to significantly strengthen our manuscript. The manuscript has been extensively revised based on your reviews. Point-by-point responses to your comments are listed below:

REVIEWER #1; EXPERT ON MTAP AND FOLATE METABOLISM:	2
REVIEWER #2; EXPERT ON CLINICAL TRIALS STATISTICS:.....	7
REVIEWER #3; EXPERT ON UROTHELIAL CLINICAL TRIALS:	11

We inserted the revised and newly added figures and contents into the letter for easy accessibility.

Figures added: Figs. 3, Supplementary Figs 3, 4, 5 and 7

Figures revised: Fig. 4

Tables revised: Table 1, Supplementary Table 2, 5

Tables added: Table 2, Supplementary Table 6

Reviewer #1; expert on MTAP and folate metabolism:

In the study by Alhalabi et al, the authors report that Pemetrexed is more effective in MTAP deficient UC than in MTAP proficient UC. They find that this hold true in a panel of UC cell lines and that depleting MTAP from a MTAP proficient line led to increased sensitivity to pemetrexed. Similarly, they found that xenografts derived from a MTAP proficient cell line were relatively insensitive to pemetrexed as compared to xenografts derived from a cell line that is MTAP deficient. Lastly, in a lung cancer clinical trial, MTAP low patients had better responses to Carboplatin plus Pemetrexed than MTAP high patients. Although there are no data indicating whether the correlation with MTAP levels has something to do with either Carboplatin or Pemetrexed.

1- The data present are clear and convincing, for the most part, and as expected. Unfortunately, blockade of de novo purine biosynthesis in MTAP deficient tumors is not novel. Neither is specifically use of pemetrexed in MTAP deficient tumors. (see: Mol Cancer Ther 2011;10:495-504; Cancer Biol Ther 2011;11:627-632; J. Clin. Oncol. 37, 385–385 (2019); Sci Rep 10, 843 (2020)).

Authors' response:

We thank the Reviewer for their comment that our data for the most part are clear and convincing. Our report is the first to thoroughly investigate blockade of de novo purine biosynthesis using pemetrexed in MTAP deficient urothelial cancer from a preclinical and clinical prospective. Our preliminary analysis, cited by the Reviewer, J. Clin. Oncol. 37, 385–385 (2019), was presented at the ASCO Genitourinary Oncology Symposium 2019 and our manuscript here provides the in-depth analysis of the earlier report. Nonetheless, we agree with the feedback provided by the Reviewer to increase the novelty of our work and have addressed it with additional experiments as detailed below.

2- A possible way to increase the novelty of the work would be to assess whether the treatment with Pemetrexed leads to increased DNA damage

Authors' response:

We appreciate the Reviewer's suggestion to increase the novelty of our work. *In vitro*, we have assessed whether pemetrexed leads to increased DNA damage by assessing the phosphorylation of histone H2AX (producing γ H2AX) and development of 53BP1 foci, both of which have been established as sensitive markers for DNA double-strand breaks (DSBs) (Bonner et al. Nat Rev Can, 2008 and Panier et al. Nat Rev Mol Cell Biol, 2014).

Our revised Methods (line 618) reflect the utilized assay. Briefly, we utilized multiplex immunofluorescence microscopy and 8 human urothelial cancer cell lines. After 24h of treatment with pemetrexed (5 μ m), we assessed the γ H2AX and 53BP1 immunofluorescent response. The 4 MTAP^{def} human urothelial cancer cell lines (253J, RT112, UMUC3 and RT4) had significantly higher γ H2AX and 53BP1 response as compared to the 4 MTAP^{prof} human urothelial cancer cell lines (T24, HT1376, HT1197 and J82). These results further support our synthetic lethality hypothesis that MTAP^{def} are more sensitive than MTAP^{prof} cell lines to pemetrexed treatment and demonstrate significantly higher DNA damage response to the same dose of pemetrexed therapy. These results (line 223) are included as a new Fig 3, which is shown for reference.

Figure 3

Figure 3

Furthermore, our *in vivo* findings demonstrate that upon treatment with pemetrexed, MTAP^{def} UM-UC-3 xenograft tumors had significantly higher 53BP1 foci as compared to the MTAP^{prof} HT-1376 tumors, indicating increased DNA damage leading to increased pemetrexed sensitivity. These results (line 251) are included as a new Supplementary Figure 7e and f, which is shown for reference.

Supplementary Figure 7e and f

3- And/or distorted nucleotide pools in patient samples and if this correlates with MTAP levels.

Authors' response:

We appreciate the Reviewer's suggestion but we, unfortunately, did not have sufficient patient samples on our trial to assess the nucleotide pools. However, we assessed the nucleotide pools *in vitro* utilizing a negative ionization method, which was developed on the Agilent 6495 LC-MS-MS to determine mono and tri-phosphate nucleotide levels. Our revised Methods (line 656) reflect the utilized assay. Measurement of triphosphate nucleotides and deoxy nucleotides did not result in consistent measurements. Therefore, we only included the measurement of nucleotide monophosphates (NMPs) in the revised manuscript.

Prior to therapy with pemetrexed, we noted that MTAP^{def} human urothelial cancer cell lines (253J, RT112, UMUC3 and RT4) had a trend for lower levels of AMP, CMP, UMP and GMP as compared to MTAP^{prof} human urothelial cancer cell lines (T24, HT1376, HT1197 and J82). However, the difference was not significant. Upon treatment with pemetrexed (5 um), NMPs increased by several folds. The increase in NMPs upon therapy trended to occur to a higher degree in MTAP^{def} cell lines (253J, RT112, UMUC3 and RT4). However, the difference was not significant likely due to the small sample size. These results (line 229) are included as a new Supplementary Figure 5a and b, which is shown for reference.

Supplementary Figure 5a and b

4- Can the authors determine, perhaps in the xenograft studies, if the MTAP deficient sensitivity to Pemetrexed has to do with accumulating MTA. Or to inhibited *de novo* purine synthesis?

Authors' response:

To address this point raised by the Reviewer, we have optimized the ultra-high performance liquid chromatography-electrospray ionization (UHPLC-ESI)-triple quadrupole mass spectrometry (described in our Methods) to measure MTA in xenograft tumors. MTA levels detected in UM-UC-3 xenograft tumor tissues were higher than HT-1376. These results are included as a new Supplementary Figure 7d, which is shown for reference.

Supplementary Figure 7d

To determine if the accumulating MTA contributes to the observed effect of pemetrexed, we used the MTT cell viability assay. We found that both MTAP^{prof} and MTAP^{def} UC cell lines were resistant to MTA therapy. Furthermore, we found that the increased sensitivity of MTAP^{def} as compared to MTAP^{prof} UC cell lines upon exposure to pemetrexed (5 μ m) was not altered with increasing doses of MTA. Based on these results and the results shown in reply to point #3, we determined that pemetrexed's sensitivity is due to the inhibited *de novo* purine synthesis. These results (line 179) are included as a new Supplementary Figure 3a and b, which is shown for reference.

Supplementary Figure 3a and b

a

b

Reviewer #2; expert on clinical trials statistics:

According to the authors, gene MTAP deficiency is significantly associated with that of the tumor suppressor gene CDKN2A in the chromosome region 9p21; they reported that patients with MTAP UC deficient had higher treatment response rate (7/11=64%) than those with MTAP UC-proficient (1/10=10%). In another independent cohort study (n=72), they found tumor MTAP-deficient was significantly associated with response rate to pemetrexed-based chemotherapy. Numerical study results are presented to justify their finding. They hypothesize that tumor MTAP deficiency may provide metabolic vulnerability for the use of antifolate agents such as pemetrexed to effectively treat UC with 9p21 loss. Thus, in study of the effects of treatment to 9p21-loss cancers, the authors propose to target the MTAP UC deficiency, and suggest this strategy to be tested in larger trials.

Major Comments:

The studies were based on MTAP-deficient and MTAP-proficient groups. I have the following comments from a biostatistician's point of view.

1 The sample size in each group is pretty small (many of them < 10), as the authors themselves acknowledged "Our study has several limitations, including the limited sample size" (p.8, line 193). This makes the authors' claim very weak. For a claim to be valid, the sample sizes of each group should be at least 100 and with significant group mean difference (p-value < 0.05).*

Authors' response:

We agree with the Reviewer's comment regarding our limited sample size. We have acknowledged this limitation in the Discussion to avoid overstating our claim. We report an observation from clinical data, which we confirmed by performing mechanistic studies. At MD Anderson, we have the opportunity to report the first and largest cohort of urothelial cancer patients to date with known MTAP status who were treated with pemetrexed. Furthermore, our lung cancer cohort serves as an additional cohort with a larger sample size given the higher prevalence of pemetrexed use in lung cancer compared to urothelial cancer.

2 The authors used the naive method (sample mean + - standard deviation, p.15, line 395-397) in their studies. This is OK if the treatment assignment (the selection of the MTAP deficient and proficient groups) is completely at random. In practice, often such assignments are not completely random, and then the naive method will give biased conclusions. In this case, the causal inference method, such as propensity score adjusted method or the doubly robust estimate method will give unbiased results. These methods are covariates adjusted and used extensively in modern clinical trials and the literature can be easily found online. I suggest the authors also use causal inference methods to validate their results, this will make their conclusion much stronger.*

Authors' response:

We apologize for our oversight regarding the labeling of our Methods subsections. The naïve method referred to in line 697 was used for the preclinical studies, which included biological duplicates in each experiment. However, our clinical data were analyzed using the statistical plan listed under line 306 and 328. The labeling in the Methods section has been revised to better convey the statistical methods for summaries and survival versus the preclinical data.

Further, we thank the Reviewer for suggesting the propensity score analysis and the doubly robust estimates, which are large sample methods. These methods do not translate to our rare samples. The multivariate methods needed to carry them out are not reliable or meaningful at these limited sample sizes, especially in the urothelial cancer cohort. Even in the larger lung adenocarcinoma cohort, only 2 degrees of freedom could be reliably used in the model, which is not meaningful for a propensity score. To address the idea that other factors may be at play, we have added baseline characteristics of the lung cohort (**Table 2**) for the reader to determine the impact other factors may contribute for future hypothesis generation. We observed that EGFR alterations were more prevalent among CDKN2A^{lo}/MTAP^{lo} as compared to the rest (42% vs 16%, P= 0.037). We discuss the clinical implication of this finding in our discussion section (line 426). These results (line 297) are included in the revised manuscript and are shown for reference.

Table 2 Baseline patient characteristics of patients with lung adenocarcinoma at the start of pemetrexed treatment

Patient Characteristics		MTAP ^{lo} /CDKN2A ^{lo}	Others
		N*(%)	N*(%)
All		26(100%)	46(100%)
Age – median (min, max)		60.5(34.0, 82.0)	59.0(26.0, 76.0)
Gender	Female	16(67%)	23(51%)
	Male	8(33%)	22(49%)
Race	Asian	1(4%)	1(2%)
	Black	1(4%)	2(4%)
	White	22(92%)	42(93%)
ECOG PS	0	1(4%)	2(4%)
	1	18(75%)	40(89%)
	2	5(21%)	3(7%)
Kras Mutation	No	17(71%)	31(69%)
	Yes	7(29%)	14(31%)
EGFR mutation	No	14(58%)	37(84%)
	Yes	10(42%)	7(16%)
Smoking Status	Current	2(8%)	6(13%)
	Former	10(42%)	20(44%)
	Never	12(50%)	19(42%)
Line of Therapy	1	16(62%)	32(70%)
	2	8(31%)	8(17%)
	3	1(4%)	5(11%)
	4	1(4%)	0(0%)
	5	0(0%)	0(0%)
	6	0(0%)	1(2%)

MTAP^{lo}: MTAP below median expression, CDKN2A^{lo}: CDKN2A below median expression, ECOG PS: Eastern Cooperative Oncology Group performance status.

* Patients with unavailable information for a specific feature were not included, so counts may not always sum to 26 and 46.

3* Often the treatment effect is affected by a group of genes, instead of a single gene such as MTAP. Although the effect of each single gene in that group is highly insignificant, but the collective group effect can be very significant. Such effect can be investigated using methods such as the rare-variants method (Morris and Zeggini, 2010; Yuan et al., 2012). The authors are encouraged to investigate this method also, possibly with collaboration of some statistician.

Authors' response:

We thank the Reviewer for suggesting the rare-variants method. In the urothelial cancer cohort, we assessed the status of the MTAP protein using immunohistochemistry and unfortunately do not have RNA or DNA data to assess the effect of other genes. Nonetheless, using the lung adenocarcinoma cohort where we have we used a generalized linear model (described in the revised Methods, line 550) estimating the odds ratio and p-value for 10 most frequently altered genes (beyond MTAP) in lung cancer. None of the assessed genes (beyond MTAP) showed a significant positive nor negative correlation with response to therapy. We have revised our Methods and Results to include this analysis. We show our methods and results below for reference.

Fig 4d.

Methods section lines 550:

“Generalized linear model:

To evaluate the association of the 10 most altered genes in lung cancer with drug response, generalized linear model (GLM) was used to estimate the odds ratio and p-value for each gene in the lung cancer cohort (n=72) independently, with 1 indicating a responder and 0 a non-responder. Genes with an odds ratio >1 (log (odds ratio) >0) and a p-value <0.05 are considered to be positively associated with responders. Genes with an odds ratio <1 (log (odds ratio) <0) and a p-value <0.05 are considered to be negatively associated with responders.”

Supplementary Table 6: generalized linear model (GLM) estimating the odds ratio and p-value for most frequently altered genes in the lung cancer cohort (n=72).

geneName	OddsRatio	coeff	cilow	cihigh	p_value	q_value
MTAP	0.392732231	- 0.934627244	0.163167375	0.875693009	0.027502371	0.302526078
PIK3CA	0.396821832	- 0.924267886	0.142228169	0.991731621	0.05927728	0.326025038
CDKN2A	0.630573777	- 0.461125117	0.267128178	1.395751323	0.26886466	0.475911601
TP53	0.948629464	- 0.052737006	0.456917359	1.979880005	0.886436851	0.886436851
ROS1	1.143344857	0.133958052	0.885756175	1.481448888	0.302852837	0.475911601
RET	1.148192416	0.138188894	0.26788913	4.591067475	0.845135547	0.886436851
MET	1.226373229	0.20406122	0.921204317	1.661766255	0.169596079	0.373111374
KRAS	1.263453703	0.233849005	0.500967174	3.241732772	0.619256902	0.85147824
ALK	1.402106297	0.337975604	0.068735069	30.3297377	0.825786859	0.886436851
EGFR	1.620237041	0.48257246	0.858411649	3.231309557	0.148775108	0.373111374
BRAF	2.317219518	0.840367983	0.725596731	8.220870683	0.167996811	0.373111374

Minor comment

4 If the authors' hypothesis/proposal on MTAP is the first, please emphasize it.*

Authors' response:

We thank the Reviewer for the comment. We have revised the Discussion section to emphasize that our hypothesis on MTAP and antifolates is the first in urothelial cancer.

Reviewer #3; expert on urothelial clinical trials:

1. Patients benefiting from pemetrexed may include those beyond responders- e.g. durable stable disease ≥ 6 months. Including them in the benefiting category may help refine the analysis?

Authors' response:

We appreciate the Reviewer's point regarding refining clinical benefit beyond responders. We have revised **Supplementary Table 2 and 5** to include patients with stable disease ≥ 6 months and those with stable disease < 6 months. The revised tables are listed below for reference.

Supplementary Table 2: Response of patients with urothelial carcinoma to pemetrexed based on tumor MTAP status

	PD	SD ≥ 6 mos	SD < 6 mos	PR	CR	NE	ORR (N)	CB (N)
MTAP ^{prof} (n=10)	5	1	3	1	0	0	10% (1)	20% (2)
MTAP ^{def} (n=11)	1	1	1	5	2	1*	64% (7)	73% (8)

MTAP^{def}, MTAP deficient; MTAP^{prof}, MTAP proficient; PD, progressive disease; SD, stable disease; mos, months; PR, partial response; CR, complete response; NE, nonevaluable, ORR: overall response rate, CB: clinical benefit. ORR was defined as PR + CR. CB was defined as SD ≥ 6 mos + CR + PR. *One patient died due to motor vehicle accident prior to evaluation with restaging scans.

In the lung cancer cohort (Supplementary Table 5), after combining patients who had SD ≥ 6 months with those who had clinical response into a clinical benefit (CB) group, we still see a trend for CDKN2A^{lo}/MTAP^{lo} patients to have higher CB (58%) than CDKN2A^{hi}/MTAP^{hi} patients (44%). However, this did not refine our analysis, suggesting that in chemotherapy treated patients, it may be better to use the traditional RECIST 1.1 criteria for response analysis and biomarker correlation. This is phenomenon is different than what's observed with immunotherapy, which can result in long term disease control without reaching CR/PR (reference: Wolchok et al, Clin Cancer Res, 2009).

Supplementary Table 5: Response of patients with lung adenocarcinoma to pemetrexed-containing chemotherapy based on CDKN2A/MTAP status by mRNA

	PD	SD ≥ 6 mos	SD < 6 mos	PR	CR	ORR (N)	CB (N)
CDKN2A ^{hi} /MTAP ^{hi} (n=25)	7	4	7	7	0	28% (7)	44% (11)
CDKN2A ^{hi} /MTAP ^{lo} (n=13)	3	4	2	4	0	31% (4)	61% (8)
CDKN2A ^{lo} /MTAP ^{hi} (n=8)	3	5	0	0	0	0% (0)	62% (5)
CDKN2A ^{lo} /MTAP ^{lo} (n=26)	5	1	6	13	1	54% (14)	58% (15)
Total (n=72)	18	14	15	24	1	35% (25)	54% (39)

PD: progressive disease, SD: stable disease, mos: months, PR: partial response, CR: complete response, ORR: overall response rate, CB: clinical benefit. OR was defined as PR + CR. CB was defined as SD ≥ 6 mos + CR + PR.

2. Authors could comment on any association with benefit from another antimetabolites-gemcitabine- which is frequently used in many cancers including urothelial and lung cancer.

Authors' response:

We appreciate the Reviewer's suggestion given the common use of gemcitabine in urothelial cancer. To test the association between MTAP deficiency and benefit from another antimetabolite chemotherapy agents, we assessed the sensitivity of urothelial cancer cell lines to gemcitabine after 48 and 72 hours of treatment. We did not observe a difference in the viability of MTAP^{def} cell lines in comparison to MTAP^{prof} cell lines. Furthermore, γ -H2AX staining showed similar effects on DNA damage were induced in both MTAP^{def} and MTAP^{prof} cells treated with 1 μ M gemcitabine for 24h. These results (line 182) are included as a new Supplementary Figure 5a and b, which is shown below for reference.

Supplementary Figure 5a and b

Furthermore, we reviewed the pathological response rates in an independent cohort of patients with localized muscle invasive urothelial cancer treated with neoadjuvant gemcitabine/cisplatin or gemcitabine/cisplatin/ifosfamide (CGI) chemotherapy (n=30). We did not find an association between response and MTAP status. These data were not included in the revised manuscript but is included here as Fig R1 to address the Reviewer's suggestion.

Fig R1.

3. Please discuss the clinical utility given that 10% of metastatic urothelial carcinoma and MTAP proficient tumors did respond to pemetrexed. A combination of markers may be more optimal?

Authors' response:

We have revised our Discussion section to include the clinical utility of pemetrexed in MTAP proficient urothelial cancer. We agree that a combination of markers might be more optimal for selection of responders. However, our cohort with metastatic urothelial cancer did not have sufficient tissue to test other markers. Our future direction of work is to optimize the biomarker and study MTAP among a combination of biomarkers that have been associated with responsiveness to pemetrexed such as TS and DHFR.

4. Genomic analyses may not always report MTAP or 9p or p16 loss? Association of MTAP loss with TMB and other actionable activating mutations may be useful to evaluate and discuss (e.g. FGFR3 in urothelial and EGFR in lung). There is also a reported sensitivity of EGFR mutated lung cancer to pemetrexed?

Authors' response:

We thank the Reviewer for their note. We have assessed patients with lung adenocarcinoma (LUAD) who have undergone targeted DNA sequencing (N=68) for association between *EGFR* mutations and *MTAP* expression. We found that patients with low *MTAP/CDKN2A* expression in their tumor were more likely to harbor an activating *EGFR* mutation as compared to patients who did not have low expression (42% vs 16%, p=0.04). We have revised our Results (line 284) and Discussion (line 408) sections to reflect our findings. Furthermore, a recent report by Bratslavsky et al in ASCO GU 2021 showed similar findings of *FGFR3* and *PTEN* activating genomic alterations being more frequent in the MTAP deficient urothelial bladder cancer. Perhaps the reason why *EGFR*-mutant lung adenocarcinoma responds well to pemetrexed is the association with MTAP loss, which is not routinely tested. This was discussed in the Results and Discussion as well.

5. Was there controlling for clinical prognostic factors in the clinical datasets, ie was there association of MTAP with pemetrexed response independent of major clinical prognostic factors, ie performance status, sites of metastasis?

Authors' response:

The Reviewer's point is well taken. Given the limited urothelial cancer sample size (N=21), we were not able to control for clinical prognostic factors when assessing response to pemetrexed. However, we did expand **table 1** to report major clinical prognostic factors such as performance status, sites of metastasis and prior lines of therapy. Furthermore, we added **table 2**, which compared the baseline prognostic features in lung adenocarcinoma cohort based on *MTAP* and *CDKN2A* expression such as age, performance status, smoking status and number of prior therapies.

6. Also, association of MTAP loss with clinical prognostic factors would be useful to report.

Authors' response:

As described in the previous point (#5), we have investigated the association of MTAP loss with clinical prognostic features in our small urothelial and lung cancer cohorts and were not able to find an association, which could be largely due to limited sample size.

References

Morris AP and Zeggini E. (2010). An evaluation of statistical approaches to rare variant analysis in genetic association studies. Genetic Epidemiology, 34: 188-193.

Yuan, A., Chen, G., Zhou Y., Bentley, A., Rotimi, C. (2012). A novel approach for the simultaneous analysis of common and rare variants in complex traits, Bioinformatics and Biology Insight. 6: 1-9.

REVIEWER COMMENTS

Reviewer #1 (Remarks to the Author):

The authors have done an admirable job addressing my concerns and suggestions. The manuscript is improved and the novelty more clear. I have no further concerns. Well done.

Reviewer #2 (Remarks to the Author):

NCOMMS-20-37775-T.R1

MTAP deficiency creates an exploitable target for antifolate therapy in 9p21-loss cancers

Submitted to: Nature Communications

The authors were unable to fully address my comments, due to the unavailability of data with larger sample at present, although they indicated a large data will be available. As a compensation, they used Fisher's exact test, which is good for small sample size. However, their claims are very weak due to the very small sample size. They acknowledged this limitation in the Discussion to avoid overstating their claim.

Their claim "metastatic MTAP-deficient UC had a higher response rate to pemetrexed (7/11=64%) than those with MTAP-proficient (MTAP-prof) UC (1/10=10%)" is premature; the claim "metastatic lung adenocarcinoma (n = 72), we found tumor MTAP-deficient was associated with a significantly improved response rate to pemetrexed-based chemotherapy" is OK.

Reviewer #3 (Remarks to the Author):

none

Re: NCOMMS-20-37775A

Title: MTAP deficiency creates an exploitable target for antifolate therapy in 9p21-loss cancers

Reviewer #1 (Remarks to the Author):

The authors have done an admirable job addressing my concerns and suggestions. The manuscript is improved and the novelty more clear. I have no further concerns. Well done.

Authors' response:

We thank the Reviewer for their comment.

Reviewer #2 (Remarks to the Author):

The authors were unable to fully address my comments, due to the unavailability of data with larger sample at present, although they indicated a large data will be available. As a compensation, they used Fisher's exact test, which is good for small sample size. However, their claims are very weak due to the very small sample size. They acknowledged this limitation in the Discussion to avoid overstating their claim.

Their claim "metastatic MTAPdef UC had a higher response rate to pemetrexed (7/11=64%) than those with MTAP-proficient(MTAPprof) UC (1/10=10%)" is premature; the claim "metastatic lung adenocarcinoma (n = 72), we found tumor MTAPdef was associated with a significantly improved response rate to pemetrexed-based chemotherapy" is OK.

Authors' response:

We appreciate the Reviewer's feedback regarding . We have modified the claim "metastatic MTAPdef UC had a higher response rate to pemetrexed (7/11=64%) than those with MTAP-proficient(MTAPprof) UC (1/10=10%)" to be "Seven enrolled patients and 14 historic controls demonstrate that MTAPdef UC have an improved ORR to pemetrexed (7/11=64%) compared to MTAP-proficient UC (1/10=10%)."

Reviewer #3 (Remarks to the Author):

None

Authors' response:

Thank you.